# Characteristics of Black Ginseng (*Panax ginseng* C.A. Mayer) Production Using Ginseng Stored at Low Temperature after Harvest

**DOI:** 10.3390/metabo11020098

**Published:** 2021-02-10

**Authors:** Hyo Bin Oh, Ji Won Lee, Da Eun Lee, Soo Chang Na, Da Eun Jeong, Dae Il Hwang, Young Soo Kim, Chung Berm Park

**Affiliations:** 1Institute of Jinan Red Ginseng, Jinan-gun 55442, Korea; tvfshin@ijrg.re.kr (J.W.L.); lee819@ijrg.re.kr (D.E.L.); nasoochang@ijrg.re.kr (S.C.N.); dmsgktn333@ijrg.re.kr (D.E.J.); dh5366@ijrg.re.kr (D.I.H.); pcbberm@ijrg.re.kr (C.B.P.); 2Department of Food Science and Technology, Jeonbuk National University, Jeonju 54896, Korea; yskim@jbnu.ac.kr

**Keywords:** benzo[a]pyrene, black ginseng, free sugar, ginseng, manufacturing, processed ginseng

## Abstract

Ginseng processing often involves multiple drying and heat treatments. Ginseng is typically processed within one week of harvesting or is stored at low temperatures to prevent spoilage. Black ginseng (BG) is manufactured by repeating the heat treatment and drying process of ginseng several times. We compared the suitability of low-temperature stored ginseng (SG) and harvested ginseng (HG) as the components for black ginseng production. SG and HG were processed into black ginseng and the appearance change, free sugar content, and benzo[*a*]pyrene (BAP) content were observed. Appearance observations showed the SG to be suitable in terms of quality when heat-treated at a temperature of 95 ℃ or higher. The BAP content of the SG increased significantly as the steaming process was repeated. A maximum BAP concentration of 5.31 ± 1.12 μg/kg was measured in SG steamed from 2 to 5 times, making it unsuitable for processing into BG. SG and HG showed similar trends in the content of sucrose, fructose, and glucose during steaming. This study aimed to facilitate the proper choice of base material to improve the safety of black ginseng by limiting BAP production during processing.

## 1. Introduction

Korean Ginseng (*Panax ginseng* C.A. Mayer) is one of the traditional medicinal plants cultivated in Korea, and it has been widely used in traditional medicines and functional foods [1]. It is typically cultivated for 4, 5, or 6 years. The bioactive saponin components of ginseng are the ginsenosides, which are classified into protopanaxadiol (PPD type) and protopanaxatriol (PPT type) [2]. Ginseng cultivation occurs between spring and autumn; if it is not consumed within one week of harvest, it will quickly decay. To prevent spoilage, it is typically stored at low temperatures or stored for a long time in the form of processed ginseng. Fresh ginseng is currently stored at low temperatures for up to 10 weeks. Research on reducing the spoilage rate of ginseng is ongoing, but changes in components such as free sugar, reducing sugar, and free amino acids during low temperature storage have not been reduced [3,4].

Ginseng is classified into white ginseng (WG), red ginseng (RG), Tae-geuk ginseng (TG), and black ginseng (BG) depending on the processing method. WG is dried in warm air or sunlight, and RG is first steamed before being dried. TG is manufactured by immersing it in hot water for a certain period of time before drying it. BG is manufactured by repeating the process of steaming and drying up to nine times [5].

Ginsenoside Rg3, ginsenoside Rk1, and ginsenoside Rg5 contents in BG are up to 30 times higher than those in RG [6,7]. Many studies have reported that ginseng has various functions, including hypoglycemic action, anti-cancer effects, anti-obesity effects, antioxidant effects, and liver function improvement [8,9]. However, through multiple steaming cycles the ginseng is carbonized. Without proper moisture during the heat treatment, benzo[*a*]pyrene (BAP), a carcinogen, is produced [10,11]. It is classified as a Group 1 carcinogen by the International Agency for Research on Cancer (IARC) [7]. In the Korean ginseng industry law, the detection standard of BAP in BG is limited to 2.0 μg/kg or less. BAP can exist in nature due to environmental pollution. It is also known to be produced by the thermal decomposition of carbohydrates, proteins, and fats, which are the main ingredients of food [12]. As consumer awareness of harmful substances increases, the safety of herbal medicines traditionally manufactured by heat treatment becomes of interest. Although many studies have been conducted on BG, studies that monitor changes in BAP content after steaming processes with varying parameters such as steaming temperature, steaming time, and steaming method are lacking. Additionally, studies looking at the use of different raw materials are insufficient.

In this study, stored ginseng (SG) and harvested ginseng (used within a week after harvesting, HG) were used to observe external changes when manufacturing BG under various conditions. In addition, by measuring contents of BAP and free sugar under various steaming conditions, the effect of free sugar on BAP production was examined to determine the potential of SG and HG as raw materials for BG production.

## 2. Results

### 2.1. Steam Processing and Appearance Observations

In this study, BG was prepared using a steaming process repeated up to five times, as shown Figure 1.

Observing the change in chromaticity of BG according to steaming temperature and time, the (redness) value tended to increase compared to WG based in the steaming process, as shown in Figure 2. This is due to the phenomenon that ginseng becomes red when dried by steam. However, after two steaming processes, the color changed from reddish brown to black and tended to become blacker over additional steaming processes. In the case of the b (yellowness), value it decreased continuously as the steaming process was repeated. In the case of the L (lightness) value, it decreased significantly for up to three steaming processes, after which it stabilized. The appearance of BG changes in proportion to the heat treatment time and temperature. This was similar to the results of measuring the chromaticity of manufactured BG in a study by Nam et al. [13]. The difference in color difference (ΔE) between SG and HG is shown in Table 1. 

The color ΔE value of SG was from 76.42 ± 0.08 to 24.19 ± 0.05 in the 95 °C 3-h steaming process, and the ΔE value of HG was from 59.84 ± 4.33 to 23.35 ± 0.11. This means that the more that the SG goes through the heat treatment process, the higher the degree of browning. In the case of steaming at 95 °C for 6 h, the L value, a value, and b value of the SG showed the greatest decreases. The difference in color difference between SG and HG during heat treatment is shown in Figure 3.

### 2.2. Free Sugar

Ginseng is known to have different levels of constituents depending on the harvest time and quality (WG, RG, and BG). Free sugar is an ingredient that changes a lot during cultivation and storage [14]. Free sugar levels tend to be the same in SG. Several types of saccharides are present in plants, which can be chemically divided into reducing sugars and non-reducing sugars. Reducing sugars are closely related to changes in the quality and browning of medicines and foods [15]. In the case of ginseng stored at low temperature for extended periods, the starch is decomposed and converted into free sugar, which is primarily the result of enzyme activity [16]. In addition, the sugars of protopanaxadiol (PPD) ginsenoside, Rb1, Rb2, Rc, and Rd, and protopanaxatriol (PPT) ginsenoside, Rg1 and Re, are removed by repeated heat treatment during the manufacture of BG, increasing the content of free sugar [17]. Browning in the heat treatment process of ginseng is an amino-carbonyl reaction, which is caused by a Maillard reaction between sugars and amino acids [18]. Components such as histidine, lysine, and arginine, whose basic amino acids are free sugars, can accelerate the reaction [19]. When analyzing the major free amino acids of red ginseng in a previous study, the content of arginine was found to be the highest, followed by asparagine and β-alanine [15]. Even in the case of BG with its relatively high free sugar content, changes in sugars during heat treatment have been observed due to the browning reaction occurring between sugars and free amino acids.

Results of free sugar analyses are shown in Table 2, WG (381.11 ± 73.97 mg/g) of SG had higher sucrose contents than that of WG (308.45 ± 4.74 mg/g) of HG.

These results show the decomposition of starch during the storage of SG [16]. The sucrose content decreased more quickly with higher temperature and longer heat treatment times. Glucose was not detected in WG in the control group, however, it generally increased during the heat treatment process. This is the same phenomenon as the glucose separation of minor ginsenoside components by heat treatment [20,21]. However, there are cases where the active ingredients are often lost due to the bursting of stored ginseng during the steaming process. This was seen with the S-T95-3h and S-T90-6h test protocols. Fructose was not detected in white ginseng and increased throughout the steaming process. Maltose was detected during the 3-h heat treatment process in the stored ginseng but not during the 6-h heat treatment process. In the case of HG, maltose content was the highest at 54.79 ± 14.64 mg/g after one steaming process, and decreased thereafter. 

In a study by Chang et al., ginseng was stored at 4 °C for up to 10 weeks with samples collected at 1-week intervals before processing into red ginseng to investigate the free sugar content [3]. In their results, fructose over 10 weeks increased from 0.47% to 4.70%. Glucose increased to 2.31% at 5 weeks and then decreased [22]. Sucrose increased to 20.55% at 3 weeks and then decreased. Maltose was measured at 6.62% before storage, which dropped to 1.37% at 10 weeks [3]. Similar results were obtained after storing ginseng for 10 weeks in the present study. In addition, SG had a higher sucrose content than that of freshly harvested ginseng by about 68.82%.

Previous studies agree with the current study, showing that sucrose content increases due to the decomposition of starch during cold storage and decreases after heat treatment [23,24]. Kim et al., reported that starch content was decreased from 17.32% to 5% after storing ginseng at 2 ± 1 °C for 20 to 30 days. Sucrose content was increased from 3.27% to 12.60% on the 20th day. In the case of maltose, none was found on the 1st day [23]. However, it reached a maximum of 2.88% on the 5th day before decreasing to 0.74% on the 20th day; these results are similar to the present study. Do et al., reported that maltose is the sugar that promotes the most browning during steaming, followed by glucose and fructose. Among the amino acids of ginseng, lysine, histidine, and arginine can significantly promote browning [15]. In the production of BG, non-enzymatic browning caused by repeated steaming is mainly due to the free sugars and amino acids reacting. The surface is then carbonized and free sugar components are converted to browning substances. After that, the rate of BAP production increases due to carbonization (rather than browning), which becomes more severe in the absence of free sugars and amino acids. In the present study, SG showed a higher conversion rate of free sugars during the steaming process than did HG. This is likely due to the decomposition of starch during low temperature storage causing an increase in free sugars while the content of free amino acids is increased by enzyme activity [24,25,26]. During heat treatment, the free amino acids and free sugars in SG react quickly, resulting in more rapid browning and carbonization than seen in HG.

### 2.3. Benzo[a]pyrene

For BG, the rate of BAP production might be increased due to a reduced water content causing deeper carbonization through repeated steaming and drying. At the same time, sucrose is hydrolyzed and decomposed into glucose and fructose. During repeated steaming cycles, glucose and fructose can react with free amino acids to produce browning or dehydrate to 5-hydroxymethylfurfural (5-HMF). 5-HMF can form a furfural derivative with one aldehyde group falling off. The benzene ring is then formed through polymerization and several rings are polymerized to form BAP [27,28,29]. The process of producing BAP from free sugar is shown in Figure 4.

As shown in Table 3, the BAP content of SG was 1.11 ± 0.57 μg/kg after one heat treatment process for three hours, whereas for HG it was 0.44 ± 0.03 μg/kg.

In the case of HG, the BAP content was 0.19 ± 0.06 μg/kg after one heat treatment process for six hours, which was lower than that of SG. SG BAP content increased to 2.92 ± 1.29 μg/kg after 3 treatments at 95 ℃ for 3 h. In addition, it increased to 4.10 ± 2.04 μg/kg after 3 treatments at 95 ℃ for 6 h. HG was divided into main roots and fine roots and BAP was measured. The main and fine roots of black ginseng were classified as shown in Figure 5. BAP content was 0.24 ± 0.04 μg/kg when heat-treated 5 times in the main roots and 0.81 ± 0.02 μg/kg in the fine roots, which were lower than that of SG.

SG steamed at 85 °C for 3 h did not show increased production of BAP. However, in industrial production, the steaming temperature is typically between 94 and 99 °C [13,14]. This is to preserve the appearance and quality by avoiding ginsenoside conversion, cracking, and internal exposure during processing [30]. In the Korean ginseng industry law, the BAP content of BG is required to be lower than 2.0 μg/kg [31]. Our results indicate that SG can be used as a raw material for RG that is heat-treated once. However, it is considered unsuitable as a raw material for BG as the BAP content is exceeded during processing. In addition, in the study of Jo et al., when the main root and the fine roots were classified and analyzed as in this study, the amount of BAP produced in the fine roots was more than four times that of the main root [9,10]. Our study results agree with these previous data showing much higher BAP production in the fine root structures, and thus, fine roots should be prepared in a separate process. In addition, other studies have reported that the optimal conditions for preparing black ginseng containing safe levels of BAP are steaming at temperatures between 80 and 120 °C and drying at temperatures below 50 °C [10]. When manufacturing black ginseng, it is recommended to use harvested ginseng in the repeated heat treatment and drying processes [9,10]. In this study, similarly, when manufacturing black ginseng, a water check of the ginseng was used, and the heat treatment temperature was 95 °C and the time was within 5 h. In addition, there is a risk of increasing the content of BAP above safe limits when repeating the steaming process more than five times. This makes quality control extremely important when processing SG.

## 3. Materials and Methods

### 3.1. Materials and Equipment

SG was created from ginseng grown for four years (with weights over 70 g), harvested in March 2019 in Jinan-gun, Jeollabuk-do, Korea, and stored for 10 weeks in a low temperature facility at −1 °C~0 °C. Freshly harvested ginseng was four-year-old ginseng (over 70 g) harvested in March 2020 in Jinan-gun, Jeollabuk-do, Korea.

Solvents used in this study included methanol (HPLC Grade, Duksan, Korea), tertiary distilled water, acetonitrile (HPLC Grade, J.T Baker, USA), ethanol (HPLC Grade, Duksan, Korea), hexane (HPLC Grade, Duksan, Korea), and dichloromethane (HPLC Grade, Duksan, Korea). Benzo[a]pyrene standards and internal standards were prepared by dissolving benzo[a]pyrene (98%, Sigma-Aldrich Co., Darmstadt, Germany) and 3-methylcholine (100 ppm, Sigma-Aldrich Co., Germany) in acetonitrile at a concentration of 1 μg/mL followed by serial dilution. Free sugar standards included glucose (99.9%, Sigma-Aldrich Co., Germany), fructose (100%, Sigma-Aldrich Co., Germany), maltose (99%, Sigma-Aldrich Co., Germany), lactose (100%, Sigma-Aldrich Co., Germany), and sucrose (99.9%, Sigma-Aldrich Co., Germany). They were dissolved in distilled water to a concentration of 1 mg/mL followed by serial dilutions.

For analyses, HPLC-DAD (Diode Array Detector (Agilent, Palo Alto, CA, USA)), FLD (Fluorometric Detector (Agilent, Palo Alto, CA, USA)), RID (Refractive Index Detector (Agilent, Palo Alto, CA, USA)), Agilent 1200 series HPLC system (Palo Alto, CA, USA), ultrasonic extraction device (JEIO TECH, UC-20, Co., Daejeon, Korea), and color difference meter (KONICA MINOLTA SENSING, Inc., Tokyo, Japan) were used. For BG manufacturing, industrial pilot steam generators (dry heat + moist heat) and drying facilities were used.

### 3.2. Steam Processing and Appearance Observation

To prevent cracking during steaming, the ginseng was dried in a drying facility at 40 °C for 6 h, left at room temperature for 12 h, and then used for the production of black ginseng. The steaming process used dry heat and wet heat for steaming. After steaming, the drying process for 2 h and the process of leaving it at room temperature for 24 h were repeated 5 times. The sample was then dried at 40 °C until the moisture content was 15% or less. The control group was dried at 40 °C without steaming to prepare white ginseng. Samples were made from three roots for each steaming condition. The surfaces of the ginseng samples were quantified using a colorimeter to measure L (Lightness), a (Redness), and b (Yellowness) values three times for each manufactured sample.

### 3.3. Sample Preparation

#### 3.3.1. Benzo[a]pyrene

Standards of BAP and 3-methylcholinerene were each dissolved in acetonitrile to a concentration of 1 μg/mL. Appropriate amounts of standard and internal standards were accurately taken and diluted with acetonitrile to prepare 3, 5, 10, 20, and 40 ng/mL of BAP and 50 ng/mL of internal standard. Approximately 5.0 g of each sample was precisely weighed. After adding 100 mL of water, ultrasonic extraction was performed for 90 min. After adding 100 mL of hexane and 1 mL of internal standard solution, the mixture was homogenized for 5 minutes followed by ultrasonic extraction for 30 min. The hexane layer was transferred to a separatory funnel. After adding 50 mL of hexane to the water layer, the mixture was shaken twice and then the hexane layer was taken and combined in the separatory funnel. After adding 50 mL of water to the hexane layer and washing, the hexane layer was dehydrated, filtered using a filter paper containing sodium sulfate, and then concentrated under reduced pressure in a water bath at 45 °C until about 2 mL of hexane was left. A florisil cartridge was used after 10 mL of dichloromethane and 20 mL of hexane were sequentially activated by addition at a rate of 2–3 drops per second. The extract was placed in the activation cartridge. Then, 20 mL of a mixture of hexane and dichloromethane (3:1) was eluted at a rate of 2–3 drops per second. The eluted solution was concentrated under nitrogen gas in a water bath at 35 °C. The residue was dissolved in 1 mL of acetonitrile and filtered through a membrane filter of 0.20 μm or less to prepare a test solution.

#### 3.3.2. Free Sugar

Each saccharide standard product (fructose, glucose, sucrose, maltose, lactose) was dried for 12 h, dissolved in distilled water, mixed, and used to prepare serial dilutions. For each test solution, 0.5 g of homogenized sample was precisely weighed. Then, 25 mL of 50% MeOH was added, refluxed at 85 °C, and cooled to room temperature. In the case of turbidity, the supernatant was centrifuged for 10 minutes and then filtered using a 0.45 μm filter to prepare the test solution.

### 3.4. Analysis of Benzo[a]pyrene by HPLC-FLD

To analyze BAP content, an Agilent 1200 series HPLC system (Palo Alto, CA, USA) equipped with a FLD detector and an Eclipse Plus C18 column (4.6 × 150 mm, 3.5 µm) was used. The injection amount was 10 μL. The flow rate was set at 1.0 mL/min. The column temperature was maintained at 30 °C. Excitation wavelength and fluorescence wavelength were set to be 294 and 404 nm, respectively, for the analysis.

A gradient elution using solvent A (water) and solvent B (acetonitrile) was used as follows: 0–30 min, 30% A and 70% B; 30–31 min, 5% A and 95% B; 31–36 min, 5% A and 95% B; 36–37 min, 30% A and 70% B; 37–45 min, 30% A and 70% B. The HPLC-FLD (Agilent, Palo Alto, CA, USA) analysis chromatogram is shown in Figure 6A.

### 3.5. Analysis of Free Sugar by HPLC-RID

To analyze free sugar content, an Agilent 1200 series HPLC system (Palo Alto, CA, USA) equipped with a RID detector was used. Agilent ZORBAX Carbohydrate 4.6 × 250 mm, 5 µm (CA, USA) was used in the column. The injection amount was 10 μL. The flow rate was 1.0 mL/min and the column temperature was 35 °C (Figure 2B). The mobile phase was analyzed for 30 minutes using water (A) and acetonitrile (B) at a ratio of 75:25 (v:v). The chromatogram of free sugar is shown in Figure 6B.

### 3.6. Statistical Analysis

For statistical analysis, a one-way distribution analysis was conducted using the SPSS 18.0 program (SPSS Inc., Chicago, IL, USA). Duncan’s multiple range tests were used to compare means between groups. Results were considered significant at *p* < 0.05 (marked by *).

## 4. Conclusions

When SG was used to prepare BG, the change in appearance was greater than that of HG. This is due to the process by which starch is decomposed and saccharified when ginseng is stored at low temperatures, which results in more rapid browning during heat treatments. Carbonization was greater in SG and the BAP content increased significantly as the number of heat treatments increased, making it unsuitable as a raw material for manufacturing BG. In the case of HG, the amount of BAP was maintained or increased only a small amount during processing. In addition, when comparing the main roots and fine roots separately, there was a slight increase in BAP in the fine roots. Therefore, we can recommend separating the main root from the fine roots before steaming and drying. It is our intention that this study informs the proper storage and use parameters in BG manufacturing in order to aid in the selection of proper raw materials and prevent unsafe products.

## Figures and Tables

**Figure 1 metabolites-11-00098-f001:**
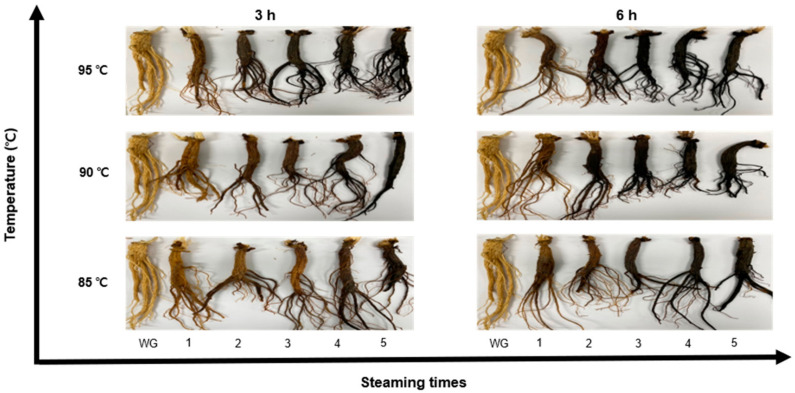
Change in the appearance of ginseng according to temperature and steaming time; WG: white ginseng.

**Figure 2 metabolites-11-00098-f002:**
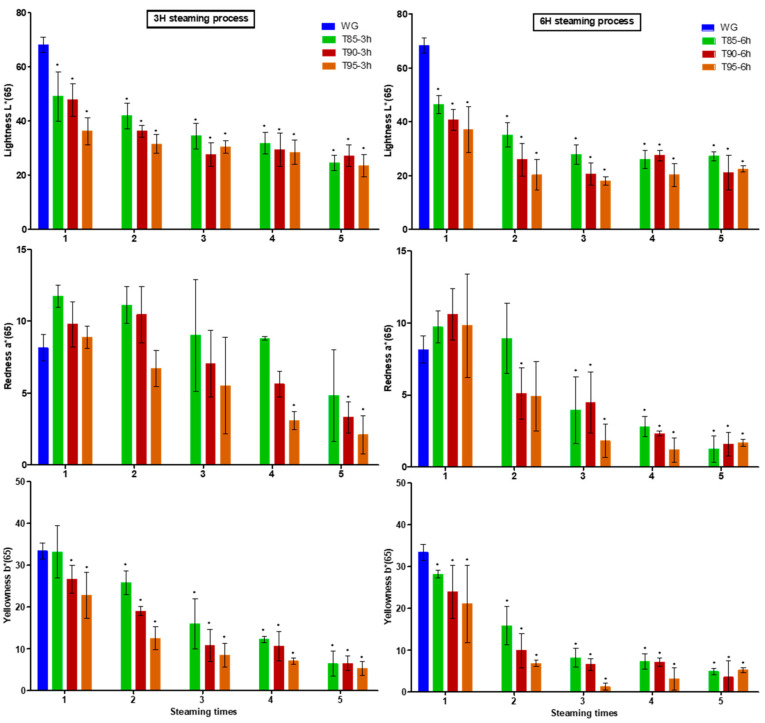
Change in appearance of black ginseng during steaming; L: Lightness; a: Redness; b: Yellowness; WG: white ginseng; T: temperature; h: steaming time. * *p* < 0.05, compared with WG.

**Figure 3 metabolites-11-00098-f003:**
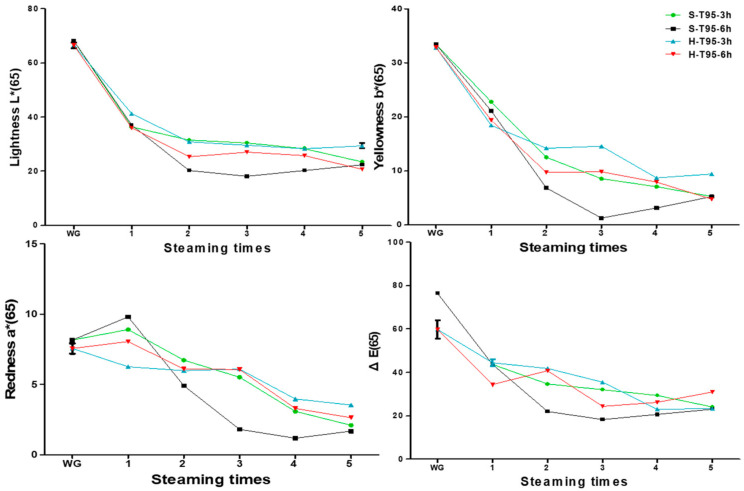
Graph of changes in color between stored and harvested ginseng based on steaming cycles; L*: Lightness; a*: Redness; b*: Yellowness; S: stored ginseng; H: harvested ginseng; T: temperature; h: steaming time.

**Figure 4 metabolites-11-00098-f004:**
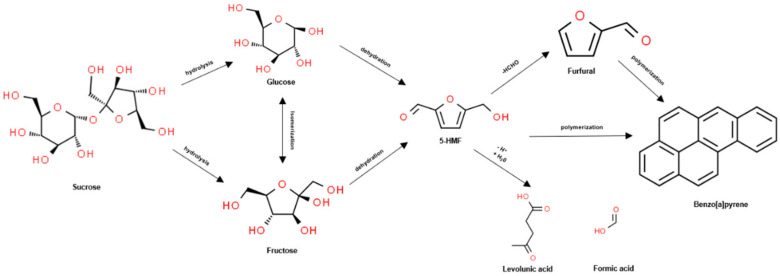
The benzo[*a*]pyrene (BAP) production process in black ginseng; 5-HMF: 5-hydroxymethylfurfural.

**Figure 5 metabolites-11-00098-f005:**
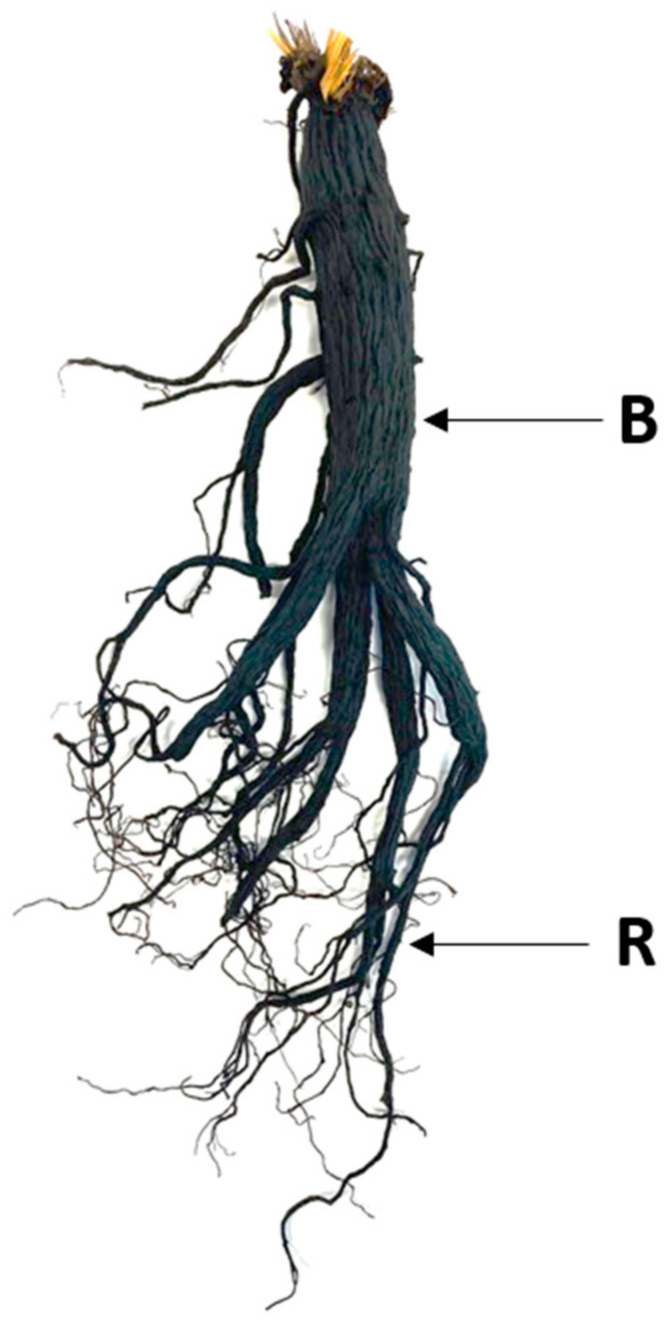
Classification of black ginseng root.**;** B: Main root; R: Fine root.

**Figure 6 metabolites-11-00098-f006:**
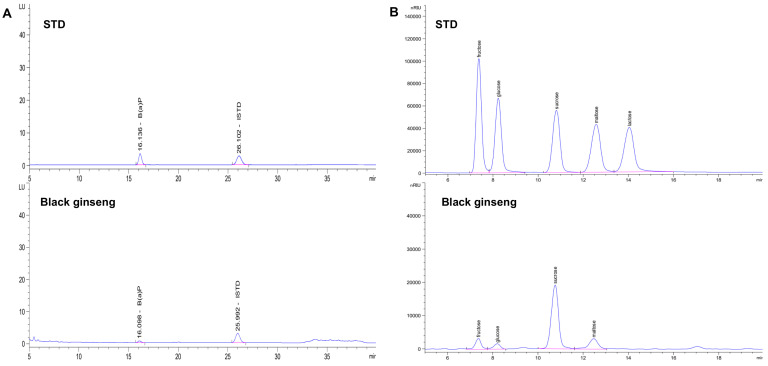
Chromatograms of HPLC-FLD, RID: (**A**) Chromatogram of benzo[*a*]pyrene; (**B**) Chromatogram of free sugar.

**Table 1 metabolites-11-00098-t001:** Change in color difference values of black ginseng by number of steaming cycles.

Steaming Process	Steaming Cycles						
		WG ^(2)^	1	2	3	4	5
S-T95-3h ^(1)^	ΔL	68.24 ± 0.06 ^(3)^	36.30 ± 0.04 *	31.53 ± 0.11 *	30.44 ± 0.20 *	28.46 ± 0.04 *	23.52 ± 0.03 *
	Δa	8.16 ± 0.02	8.91 ± 0.02	6.73 ± 0.02	5.52 ± 0.01	3.08 ± 0.01 *	2.10 ± 0.02 *
	Δb	33.42 ± 0.04	22.77 ± 0.02	12.51 ± 0.03	8.55 ± 0.03	7.07 ± 0.01	5.26 ± 0.03
	ΔE *	76.42 ± 0.08	43.77 ± 0.04	34.58 ± 0.11	32.10 ± 0.21	29.49 ± 0.04	24.19 ± 0.05
S-T95-6h	ΔL	68.24 ± 0.06	37.09 ± 0.12 *	20.31 ± 0.14 *	18.11 ± 0.18 *	20.26 ± 0.02 *	22.49 ± 0.06 *
	Δa	8.16 ± 0.02	9.81 ± 0.01	4.91 ± 0.04	1.81 ± 0.02 *	1.18 ± 0.01 *	1.68 ± 0.07 *
	Δb	33.42 ± 0.04	21.09 ± 0.05 *	6.85 ± 0.06 *	1.26 ± 0.02 *	3.13 ± 0.01 *	5.24 ± 0.15 *
	ΔE	76.42 ± 0.08	43.78 ± 0.13	21.98 ± 0.16	18.25 ± 0.18	20.53 ± 0.03	23.15 ± 0.18
H-T95-3h	ΔL	52.63 ± 2.68	37.27 ± 1.07 *	37.39 ± 0.62 *	30.98 ± 0.04 *	21.35 ± 0.23 *	21.92 ± 0.12 *
	Δa	9.51 ± 2.02	6.19 ± 0.06 *	5.54 ± 0.11 *	7.51 ± 0.01 *	3.43 ± 0.02 *	3.90 ± 0.01 *
	Δb	26.74 ± 3.85	23.51 ± 0.49 *	17.95 ± 0.14 *	15.41 ± 0.01 *	7.90 ± 0.05 *	7.04 ± 0.01 *
	ΔE	59.84 ± 4.33	44.50 ± 1.64	41.84 ± 0.48	35.41 ± 0.04	23.02 ± 0.23	23.35 ± 0.11
H-T95-6h	ΔL	52.63 ± 2.68	28.90 ± 0.14 *	35.23 ± 0.36 *	22.93 ± 0.40 *	24.67 ± 0.36 *	28.08 ± 0.16 *
	Δa	9.51 ± 2.02	6.16 ± 0.03 *	7.45 ± 0.10 *	3.61 ± 0.14 *	3.99 ± 0.13 *	7.77 ± 0.19 *
	Δb	26.74 ± 3.85	17.45 ± 0.17 *	19.12 ± 0.17 *	7.30 ± 0.15 *	8.30 ± 0.27 *	10.80 ± 0.30 *
	ΔE	59.84 ± 4.33	34.32 ± 0.13	40.78 ± 0.13	24.33 ± 0.34	26.33 ± 0.44	31.07 ± 0.29

L: Lightness; a: Redness; b: Yellowness; ΔE: $\sqrt{{\left(\Delta L\right)}^{2}+{\left(\Delta a\right)}^{2}+{\left(\Delta b\right)}^{2}}$; ^(1)^ S: Stored ginseng; H: harvested ginseng; T: temperature; h: steaming time; ^(2)^ WG: white ginseng; ^(3)^ The data are expressed as the mean ± SD (*n* = 3); * *p* < 0.05, compared with WG.

**Table 2 metabolites-11-00098-t002:** Free sugar content of stored and harvested ginseng.

Steaming Process	Steaming Times	Sucrose (mg/g)	Glucose (mg/g)	Fructose (mg/g)	Maltose (mg/g)
Control	S-WG ^(2)^	^(3)^ 381.11 ± 73.97	N.D	N.D	N.D
	M-WG	308.45 ± 4.74	N.D	9.41 ± 4.03	N.D
S-T85-3h	1 3 5	302.02 ± 35.69 261.88 ± 62.90 224.30 ± 8.95	8.81 ± 2.87 4.09 ± 3.25 17.22 ± 7.76	17.00 ± 5.48 25.24 ± 5.34 57.73 ± 9.63	3.01 ± 2.31 N.D N.D
S-T90-3h ^(1)^	1	301.55 ± 3.52 *	2.24 ± 5.87	11.04 ± 1.82 *	18.40 ± 5.64
	3	258.35 ± 3.04 *	5.06 ± 1.53 *	37.07 ± 2.16 *	N.D
	5	148.87 ± 4.85 *	38.67 ± 2.80 *	92.02 ± 1.63 *	N.D
S-T95-3h	1	283.05 ± 74.52 *	15.17 ± 6.90	29.24 ± 8.64 *	23.07 ± 3.95 *
	3	146.87 ± 5.23 *	27.74 ± 11.81 *	85.86 ± 1.53 *	6.50 ± 11.72 *
	5	23.49 ± 1.87 *	9.20 ± 7.10 *	72.86 ± 1.08 *	N.D
S-T85-6h	1 3 5	300.57 ± 17.82 197.90 ± 12.03 106.14±11.50	5.65 ± 2.04 6.01 ± 1.25 20.05 ± 7.89	14.45 ± 2.54 53.66 ± 8.62 85.13 ± 7.33	8.12 ± 3.69 N.D N.D
S-T90-6h	1	301.68 ± 17.40 *	2.08 ± 0.54	25.01 ± 8.16 *	N.D
	3	160.79 ± 8.79 *	24.46 ± 8.06 *	69.71 ± 4.61 *	N.D
	5	27.06 ± 2.99 *	12.65 ± 3.28 *	96.56 ± 2.17 *	N.D
S-T95-6h	1	258.77 ± 6.28 *	7.75 ± 6.88	25.78 ± 6.72 *	N.D
	3	46.17 ± 3.76 *	14.50 ± 5.44	86.80 ± 5.76 *	N.D
	5	N.D	35.10 ± 4.77 *	88.26 ± 1.98 *	N.D
H-T95-6h^1)^	1	193.72 ± 60.92 *	15.32 ± 5.09	27.11 ± 7.75	54.79 ± 14.64 *
	3	158.93 ± 51.37 *	34.63 ± 12.75	59.03 ± 15.14	44.90 ± 13.84 *
	5	35.79 ± 15.37 *	63.22 ± 28.70 *	106.72 ± 35.80 *	33.15 ± 22.17 *

^(1)^ S: Stored ginseng; H: harvested ginseng; T: Temperature; h: Steaming time; ^(2)^ WG: White ginseng; ^(3)^ The data are expressed as the mean ± SD (*n* = 3). The data are expressed as the mean ± SD (*n* = 3); * *p* < 0.05, compared with WG.

**Table 3 metabolites-11-00098-t003:** Benzo[*a*]pyrene content of black ginseng by steaming process. Values are expressed as μg/kg.

Steaming Process	Steaming Cycles				
	1	2	3	4	5
^(1)^ S-T85-3h	^(2)^ 0.50 ± 0.18	0.51 ± 0.17	0.51 ± 0.07	0.63 ± 0.12	0.58 ± 0.08
S-T85-6h	1.11 ± 0.57	4.80 ± 1.41 *	4.63 ± 1.49 *	5.31 ± 1.12 *	5.17 ± 1.05 *
S-T90-3h	0.61 ± 0.09	1.22 ± 0.27 *	1.56 ± 0.48 *	1.00 ± 0.40	1.72 ± 0.10 *
S-T90-6h	1.09 ± 0.14	1.50 ± 0.23	5.01 ± 1.07 *	4.62 ± 2.16 *	1.89 ± 1.58
S-T95-3h	0.93 ± 0.24	1.52 ± 0.41	1.01 ± 0.78	2.92 ± 1.29	2.58 ± 1.29
S-T95-6h	0.78 ± 0.33	3.10 ± 2.09	2.39 ± 0.49	4.10 ± 2.04 *	3.75 ± 1.21 *
H-T95-6h-B	0.19 ± 0.06	0.26 ± 0.04	0.13 ± 0.03	0.19 ± 0.04	0.24 ± 0.04
H-T95-6h-R	0.44 ± 0.03	0.75 ± 0.08	0.69 ± 0.06	0.82 ± 0.09	0.81 ± 0.02

^(1)^ S: Stored ginseng; H: Harvested ginseng; T: Temperature; h: Steaming time; B: main root; R: fine root; ^(2)^ The data are expressed as the mean ± SD (*n* = 3). * *p* < 0.05, compared with WG.

## Data Availability

The data presented in this study are available on request from the corresponding author. The data is not publicly available because it is a part of an ongoing study.

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
