# Peer review of "Characteristics of Black Ginseng (Panax ginseng C.A. Mayer) Production Using Ginseng Stored at Low Temperature after Harvest"

_metabolites, 2021, doi:10.3390/metabo11020098_

Round 1

Reviewer 1 Report

The correct name of the taxon is Panax ginseng C.A.Mey., the correct term is benzo[a]pyrene

The work should mention the origin of the ginseng plants used (variety, chemical race, etc.), as it is not clear to what extent it is a homogeneous material.

The introductory chapter should be preceded by abbreviations used in the manuscript for clarity and simplification for readers.

It is not very understandable why it is in tab. 2 lactose, although not determined and is not found in ginseng.

In the methodological part, there is nothing to indicate whether all quantitative values are related to plant samples' dry matter. This must always be stated. If this is not the case, a recalculation must be performer,

Figure. 2 - the asterisks above the columns in the graphs are not explained,

Table 1 - confusing abbreviations S: Stored ginseng; M: Harvested (formerly SG and HG). I propose revisions 1) and 3) directly in the table; are they well located?

Figure 3 - I am afraid that at least the graph concerning ΔE does not correspond to the values of M-T95-6H in Table 1,

Table 2 - the text does not explain why the changes in sugars content at 85 °C were not monitored, and in addition, the free sugars content in freshly harvested ginseng is monitored only after steaming at 95 °C and no other temperatures and times are given below,

line 170-171 - does the wording indicate that at least one steaming is not very suitable from the point of view of BAP content?

Line 175 - it would be appropriate to specify in the text that by freshly harvested ginseng, the longest steaming interval was used at the highest experimental temperature. When a low BAP content was observed in contradiction to the shortest steaming at the lowest observed temperature of stored ginseng, the meaning of the other two sentences somewhat escaped me.

line 184: I consider it appropriate to discuss the content of BAP more widely using recent publications (e.g. doi: 10.3390 / molecules24101856 etc.)

Author Response

1. The correct name of the taxon is Panax ginseng C.A.Mey., the correct term is benzo[a]pyrene

Answer: Line 2 title panax ginseng C.A. Mayer modified

Answer: Line 17, 49, 164 benzo[a]pyrene modified

2. The work should mention the origin of the ginseng plants used (variety, chemical race, etc.), as it is not clear to what extent it is a homogeneous material.

Answer: Information on the origin of ginseng is indicated in Korean ginseng.

3. The introductory chapter should be preceded by abbreviations used in the manuscript for clarity and simplification for readers.

Answer: Unfortunately, there is no abbreviation in the journal form.

4. It is not very understandable why it is in tab. 2 lactose, although not determined and is not found in ginseng.

Answer: lactose results deleted

5. In the methodological part, there is nothing to indicate whether all quantitative values are related to plant samples' dry matter. This must always be stated. If this is not the case, a recalculation must be performer.

Answer: All black ginseng samples were completely dried with a moisture content of 15% or less.

6. Figure. 2 - the asterisks above the columns in the graphs are not explained,

Answer: This asterisk has been modified to display statistics.

7. Table 1 - confusing abbreviations S: Stored ginseng; M: Harvested (formerly SG and HG). I propose revisions 1) and 3) directly in the table; are they well located?

Answer: Changed to avoid confusion of words (H: Harvested ginseng, h : Steaming time)

8. Figure 3 - I am afraid that at least the graph concerning ΔE does not correspond to the values of M-T95-6H in Table 1.

Answer: There was an error in reflecting the numerical value for ΔE. Immediately corrected.

9. Table 2 - the text does not explain why the changes in sugars content at 85 °C were not monitored, and in addition, the free sugars content in freshly harvested ginseng is monitored only after steaming at 95 °C and no other temperatures and times are given below,

Answer: Added a section on the content of free sugar in the 85 °C process. The harversted ginseng was applied at the temperature used in the industry. Harvested ginseng was applied to compare with stored ginseng using the temperature used in the industry.

10. line 170-171 - does the wording indicate that at least one steaming is not very suitable from the point of view of BAP content?

Answer: This sentence is subject to confusing and has been changed.

11. Line 175 - it would be appropriate to specify in the text that by freshly harvested ginseng, the longest steaming interval was used at the highest experimental temperature. When a low BAP content was observed in contradiction to the shortest steaming at the lowest observed temperature of stored ginseng, the meaning of the other two sentences somewhat escaped me.

Answer: Added consideration of benzo[a]pyrene. Stored ginseng is saccharified in a low temperature process and there is a high risk of benzopyrene formation during heat treatment.

12. line 184: I consider it appropriate to discuss the content of BAP more widely using recent publications (e.g. doi: 10.3390 / molecules24101856 etc.)

Answer: Added consideration of benzo[a]pyrene.

Reviewer 2 Report

The manuscript titled "Characteristics of Black Ginseng (panax ginseng C.A Mayer) Production Using Ginseng Stored at Low Temperature After Harvest" describes an interesting and important studies about limiting of benzo[a]pyrene production during processing of black ginseng. The Manuscript in overall is written correctly and content many efforts done by authors with time consuming during experiments, nevertheless in my opinion may be published after some modifications as follows:

1) Title: please insert "dot" after "A" in C.A. Mayer

2) Abstract: please provide explanation shortcut BG as you are using first time. 

3) Keywords: Please uniform capital letters in Ginseng word

4) Introduction: please uniform - capital letter "Panax"; 

Also please double check numbers of references as it seems #10 is missing. 

5) Results: please uniform references style - line 155 vs. line 165

6) Results: line 166 and line 167 - sentence "The process of producing benzopyrene..." is repeated

7) REsults: table 3 - please provide units (ppb?)

8) Materials and methods: line 210-213 - despite later you provided more information about equipment used, please be more precise and provide manufacture name of FLD and RID

8) Materials and methods: line 266 and line 273 - please provide concentration of samples injected into instrument as shown in chromatograms 

9) references: please uniform and provide references according journal requirements - ref. # 10, #11, #13, #17, #19 etc. Please provide proper journal abbreviations, proper surnames with initial of names of authors, proper and uniform references journal numbers and pages.  

Author Response

1. Title: please insert "dot" after "A" in C.A. Mayer

Answer: line 2 modified

2. Abstract: please provide explanation shortcut BG as you are using first time. 

Answer: line 13 Added description of black ginseng.

3. Keywords: Please uniform capital letters in Ginseng word

Answer: line 24 modified

4. Introduction: please uniform - capital letter "Panax"; Also please double check numbers of references as it seems #10 is missing.

Answer: line 28 modifed

5. Results: please uniform references style - line 155 vs. line 165

Answer: references style modified

6. Results: line 166 and line 167 - sentence "The process of producing benzopyrene..." is repeated

Answer: Deleted repeated sentences.

7. REsults: table 3 - please provide units (ppb?)

Answer: The unit was changed to μg/kg and reflected.

8. Materials and methods: line 210-213 - despite later you provided more information about equipment used, please be more precise and provide manufacture name of FLD and RID

Answer: added manufacture name

9. Materials and methods: line 266 and line 273 - please provide concentration of samples injected into instrument as shown in chromatograms

Answer: The injection volume of each HPLC instrument sample was 10 μL. line 273, 287

10. references: please uniform and provide references according journal requirements - ref. # 10, #11, #13, #17, #19 etc. Please provide proper journal abbreviations, proper surnames with initial of names of authors, proper and uniform references journal numbers and pages.

Answer: Modified according to journal requirements.

Reviewer 3 Report

I do not feel that Metabolites is appropiate, as this is a very limited targeted metabolomics study and would rather recommend a different journal.

Author Response

Thank you for reviewing my thesis.

Reviewer 4 Report

The manuscript from Bin Oh et al. on the effect of storage and process for black ginseng production in regards to BPA production and sugars content.

The manuscript is easy to read, I have few comments to it.

Introduction

Lines 46:47. It would be good to state which are the safety limits for BPA content in food according to the legislation.

Line 51:55. The authors mentioned that here are other studies but no references and no brief description on what the findings from other studies are

Results

Table 1. why there are only values reported for the 95 C? how about the other temperatures? Furthermore, why changing the abbreviation for the harvest ginseng from HG to MG? 

Line 88: I could not find the value 74.62 in the table 1.

If figure 3 containing the results from Table 1? It seems so and if that is the case I would keep the figure 3 which is more intuitive and having the table 1 in the supplementary material.

Table 2. Similar comment for changing the abbreviation from HG to MG. The glucose levels seems not following a trend for the 95C-3H and 90-6H for storage. Do you know why they go up and down when looking at the steaming cycles?

Line 189-190. Where can we see in your results the difference between fine roots and the main root? it is not clear to me...

Line 166:167. The sentence is repeated twice.

Line 184: It is missing the reference to the legislation regarding the limit of BPA.

Author Response

1. Lines 46:47. It would be good to state which are the safety limits for BPA content in food according to the legislation.

Answer: Added as the content of the Korean Ginseng Industry Act.

2. Table 1. why there are only values reported for the 95 C? how about the other temperatures? Furthermore, why changing the abbreviation for the harvest ginseng from HG to MG?

Answer: The pattern of decreasing with temperature was the same. Since the main purpose is to compare stored ginseng and harvested ginseng, it is presented at the same temperature and time. Changed to avoid confusion of words (H: Harvested ginseng, h : Steaming time)

3. Line 88: I could not find the value 74.62 in the table 1.

Answer: There was an error in the data notation. Immediately corrected.

4. If figure 3 containing the results from Table 1? It seems so and if that is the case I would keep the figure 3 which is more intuitive and having the table 1 in the supplementary material.

Answer: Fig. 3 suggests that the appearance of stored ginseng is changing significantly. Compared to that, harvested Ginseng shows less change and less change on the surface.

5. Table 2. Similar comment for changing the abbreviation from HG to MG. The glucose levels seems not following a trend for the 95C-3H and 90-6H for storage. Do you know why they go up and down when looking at the steaming cycles?

Answer: Stored ginseng is a phenomenon that cracks during heat treatment, and active ingredients are often leaked. This is the case.

6. Line 189-190. Where can we see in your results the difference between fine roots and the main root? it is not clear to me...

Answer: Added figures of root classification.

7. Line 166:167. The sentence is repeated twice.

Answer: Immediately corrected.

8. Line 184: It is missing the reference to the legislation regarding the limit of BPA. 

Answer: The Korean Ginseng Industry Act has been added as reference #32.